# Efficacy and Safety of Kudzu Flower–Mandarin Peel on Hot Flashes and Bone Markers in Women during the Menopausal Transition: A Randomized Controlled Trial

**DOI:** 10.3390/nu12113237

**Published:** 2020-10-22

**Authors:** Ji Eon Kim, Hyeyun Jeong, Soohee Hur, Junho Lee, Oran Kwon

**Affiliations:** 1Department of Nutritional Science and Food Management, Ewha Womans University, Seoul 03760, Korea; wldjs0903@naver.com (J.E.K.); jhy1630@hanmail.net (H.J.); soohee1276@naver.com (S.H.); 2LG Household and Healthcare Research Park, Seoul 07795, Korea; leejh222@lgcare.com

**Keywords:** *Pueraria thomsonii*, *Citrus unshiu*, menopause, hot flash, bone resorption

## Abstract

This randomized controlled study aimed to assess the efficacy and safety of an extract mixture of kudzu flower and mandarin peel (KM) on hot flashes (HFs) and markers of bone turnover in women during the menopausal transition. Healthy women aged 45–60 years with the menopausal HFs were randomly assigned in a 1:1 ratio to either KM (1150 mg/day) or placebo arms for 12 weeks (*n* = 84). The intent-to-treat analysis found that compared with the placebo, the KM significantly attenuated HF scores (*p* = 0.041) and HF severities (*p* < 0.001), with a mean difference from baseline to week 12. The KM also improved bone turnover markers, showing a significant reduction in bone resorption CTx (*p* = 0.027) and a tendency of increasing bone formation OC relative to the placebo. No serious adverse events and hormonal changes were observed in both groups. These findings suggest that KM consumption may improve the quality of life in ways that are important to symptomatic menopausal women.

## 1. Introduction

Menopause is a natural process of women aging that increases vulnerability to physical [1] and emotional [2] stresses, leading to a reduced quality of life and increased burden for health care needs [3,4]. From an endocrinological perspective, menopause is characterized by increased follicle-stimulating hormone (FSH) levels and a fluctuating/eventual decline in estrogen [5]. At some time through the menopausal transition from peri- to post-menopausal state, >80% of women will experience the vasomotor symptoms (VMS) that are commonly called hot flashes or flushes (HFs) and night sweats [6].VMS peak in late perimenopause or early menopause due to the changes in circulating estrogen levels [7]. Meanwhile, because circulating estrogen plays a vital role in preventing bone loss and promoting bone formation, menopausal estrogen deficiency may also contribute to diminished bone mineral density (BMD) [8]. In a retrospective study, Crandall et al. [9] noted that lumbar BMD was lower in perimenopausal women with frequent HFs than in non-flushing women, supporting the notion that HFs may serve as an independent determinant of BMD. In contrast, Tuomikoeki et al. [10] suggested that HFs do not appear to determine lumbar and hip BMD in a prospective longitudinal study of 143 healthy women with or without HFs.

Hormone therapy has long been recognized as a standard treatment to reduce unpleasant symptoms of menopause. However, because of the risks of breast cancer, endometrial cancer, and venous thromboembolism [11], many women now want to avoid hormone therapy. Instead, various plant food products with estrogenic activity (phytoestrogens) are widely thought to confer health benefits to menopausal women. Globally, the phytoestrogen market was estimated to be valued at USD 1.2 billion in 2019, with a compound annual growth rate of 4.7% [12]. However, the prominent dietary and herbal sources of phytoestrogens are only partially known to be safe, effective, and well-tolerated by symptomatic menopausal women in controlled clinical trials. Therefore, there is still an unmet need for safe and effective non-hormone treatment options for menopausal women.

Kudzu (*Pueraria thomsonii* Benth.) belongs to the Fabaceae/Leguminosae family. The flower of kudzu, also called the Pueraria flower, contains tectorigenin, tectoridin, and tectorigenin 7-*O*-xylosylglucoside that have lower binding affinities to estrogen receptors (ERs) than those of soy isoflavones [13]. Mandarin (*Citrus unshiu* Markovich) is a popular citrus fruit belonging to the Rosaceae family. Dried mandarin peel is rich in flavonoids, such as hesperidin, naringin, and narirutin [14]. Current research has identified the modulatory effects of mandarin peel on bone metabolism [15] and inhibitory effects of hesperidin against bone loss in ovariectomized (OVX) animals [16]. Two recent animal studies focused on the HFs and BMD [17,18], respectively, in OVX mice treated with an extract mixture of kudzu flower and mandarin peel (KM) at the ratio of 6.5:5. The first study implied that a 7-week treatment with KM exerted protective effects against HFs by enhancing serotonin, norepinephrine, dopamine, and ERβ in the hypothalamus, and decreasing circulating FSH and luteinizing hormone (LH) levels without changing the circulating estrogen level [17]. In the second study, KM demonstrated anti-osteoporotic effects after a 7-week treatment by maintaining bone homeostasis via re-addressing the balance between bone resorption and bone formation [18]. However, to the best of our knowledge, KM’s protective attributes have not yet been confirmed in human studies. It triggered us to study the potentials of KM in human subjects.

Based on these findings, we hypothesized that KM might potentially modify HFs and BMDs in menopausal women. To test this hypothesis, we conducted a double-blind placebo-controlled human trial to determine the safety and effectiveness of KM on modulating HFs and bone turnover markers in peri- and post-menopausal women.

## 2. Materials and Methods

### 2.1. Test Materials

The mandarin peel and kudzu flower are listed as Old Dietary Ingredients marketed in the US before 15 October 1994 [19]. Therefore, they might be used legally in a dietary supplement without submitting the Food and Drug Administration notification. The test and placebo products were provided by the LG Household & Health Care (Seoul, Korea). In brief, the 70% ethanolic extract of kudzu flower and aqueous extract of mandarin peel were spray-dried separately and mixed at the ratio of 6.5:5. The KM, lactose, crystalline cellulose, sodium carboxymethyl starch, silicon dioxide, magnesium stearate, and cacao color were packed in a capsule to provide a dose of 383.4 mg KM. For the placebo capsule, KM was replaced with an equal weight of lactose. The test product was standardized to contain 70.3 mg/g of three tectorigenin derivatives from the kudzu flower and 51.7 mg/g hesperidin from mandarin peel using a high-performance liquid chromatograph equipped with a variable wavelength detector (Agilent Technologies, Santa Clara, CA, USA) and Poroshell EC-C18 column (2.1 mm × 1000 mm, 2.7 μm).

### 2.2. Subjects

The required sample size was estimated based on data from Chang et al. [20] and Kim et al. [20,21] and conducted assuming a power of 0.80 with a two-sided α-level of 0.05 and 20% dropout rate. Subjects were recruited through posters placed in various locations and online advertisements. For inclusion, subjects were required to be peri- or post-menopausal women aged between 45 and 60 years with the menopausal HFs as indicated by a Kupperman Index (KI) ≥15 and average daily HF score ≥10 for 1 week before the screening visit. Exclusion criteria were as follows: (1) surgical or chemotherapy-induced menopause or unexplained vaginal bleeding; (2) receiving hormone therapy within the preceding 6 months; (3) taking a supplement or traditional medicine affecting menopausal status, bone health, blood glucose, blood lipids, blood pressure, and blood circulation within the preceding 1 month; (4) taking medication within the past 1 month or during the study period; (5) a history of breast cancer, endometrial hyperplasia, uterine endometrial cancer, sex steroid-dependent organ tumors, or having abnormal findings on breast X-ray or pelvic ultrasonography; (6) having liver or kidney diseases, uncontrolled hypertension (systolic blood pressure >160 mmHg or diastolic blood pressure >100 mmHg), thyroid disease, diabetes, hyperlipidemia, or mental illness; (7) a history of severe migraine headache, thromboembolic disorders, cerebrovascular disorders, or serious cardiovascular condition within the preceding 1 year; (8) drug addiction or alcoholism; (9) performing heavy exercise (≥10 h/week) within the past 3 months; (10) known allergy or sensitivity to ingredients in the study products; (11) enrollment in any other clinical trial within the preceding 1 month; (12) investigator’s determination of unsuitability for the trial. All participants provided written informed consent, and the Institutional Review Board of Ewha Womans University approved the protocol (IRB No. 164-14). The study was registered on the International Clinical Trials Registry Platform of the WHO (ICTRP) under the identification number KCT0003335.

### 2.3. Study Design

The study followed a randomized controlled parallel design with two arms of a fixed-dose of KM (1150 mg) or matching placebo in a ratio of 1:1. A total of 84 eligible subjects were randomized to the KM (*n* = 42) or placebo (*n* = 42) groups using a computer-generated random block number table. The group allocation was blinded for both the investigators and participants. The subjects were advised to take three test material caps with enough water once a day for 12 weeks. We determined the daily dose based on previous animal studies [18,22] and the duration by some previous studies [23]. During the trial, subjects were instructed to maintain their usual diet and lifestyle but refrain from eating or drinking kudzu, kudzu flower, mandarin peel, citrus fruits, calcium, and isoflavone-rich foods and beverages. Dietary intakes and responses to the International Physical Activity Questionnaire (IPAQ) [24] were recorded for 3 days (two weekdays and one weekend day) at baseline and weeks 6 and 12, using a smartphone application. Pittsburgh Sleep Quality Index (PSQI) [25] and Recommended Food Scores (RFS) [26] were measured at baseline and week 12 to assess the qualities of sleep and food intake, respectively. Blood samples were taken at baseline and week 12, after an overnight fast.

### 2.4. Measurement of HFs

Participants were required to complete a daily diary over 12 weeks from 7 days before treatment at baseline using a Web-based prospective electronic digital HF diary. The data were collected by the Google form survey method using a designed questionnaire in which participants recorded both the frequency and severity of HFs. For the frequency of HF, the daily number of episodes is recorded as one point. The severity of HF was defined as mild = 1, moderate = 2, severe = 3, and very severe = 4. The HF scores were calculated by multiplying the frequency experienced at that severity level. Then, the resulting points were summed to obtain a total daily score. Each daily score was averaged per week [27,28]. Daytime and night-time HFs were considered separately.

The frequency and severity of menopausal symptoms were also measured via the KI and Menopause-Specific Quality of Life Questionnaire (MenQOL) at baseline and end of the study. KI is a numerical index that scores 11 menopausal-related symptoms (HFs, paresthesia, insomnia, nervousness, melancholia, vertigo, weakness, arthralgia, myalgia, headache, and palpitations). Each symptom was rated from 0 to 3 according to severity and symptoms, weighted, and the total sum was calculated. MenQOL is a menopause-specific tool to measure health-related quality of life. It consists of a total of 29 items in a Likert-scale format, assessing the impacts on the four domains, including vasomotor (1–3 items), psychosocial (4–10 items), physical (11–26 items), and sexual (27–29 items) [29]. Means were computed by dividing the sum of the domain’s items by the number of items within that domain.

### 2.5. Measurement of Biochemical Markers in the Blood

At baseline and week 12, venous blood was collected in the ethylenediaminetetraacetic acid tube (BD Biosciences, San Jose, CA, USA) and serum-separated tube (BD Biosciences). The plasma was separated by centrifugation at 1500× *g*, 4 °C for 10 min, and serum was centrifuged at 1910× *g*, 4 °C, for 15 min. Serum osteocalcin (OC) was measured by electrochemiluminescence (Elecsys N-MID Osteocalcin ELISA Kit, Roche Diagnostics GmbH, Mannheim, Germany). Plasma C-telopeptide fragment (CTx), plasma N-telopeptide fragment (NTx), and serum bone-specific alkaline phosphatase (BALP) were assessed by the Elecsys β-CrossLaps/serum assay (Roche Diagnostics GmbH), enzyme-linked immunosorbent assay kit (Cusabio Biotech, Wuhan, China), and the Access Ostase assay (Beckman Coulter, Fullerton, CA, USA), respectively. Estradiol, FSH, and LH levels were determined by the Cobas e801 analyzer (Roche Diagnostics GmbH).

### 2.6. Safety Measurements

All participants were examined in the Hanaro Medical Foundation (Seoul, Korea) or Dong-A Radiology Clinics (Daejeon, Korea) for adverse events and side effects. Safety monitoring, including vital signs (blood pressures, pulse rate, body temperature), was carried out on every visit. Laboratory tests were conducted at baseline and week 12 for hematologic (WBC, RBC, Hb, Hct, PLT, neutrophils, eosinophils, basophils, lymphocytes, monocytes), blood biochemical (AST, ALT, ALP, BUN, creatinine, e-GFR), and urine analysis (pH, nitrite, specific gravity, protein, glucose, ketone, bilirubin, blood, urobilinogen, color). The endometrial thickness was determined by transvaginal or abdominal ultrasonography at baseline and week 12 to monitor side effects. Adverse events were monitored throughout the study.

### 2.7. Statistical Analysis

Data analyses were performed using the intention-to-treat (ITT) analysis and tested for normal distribution graphically by evaluating quantile-quantile (QQ) plots. Values that exceed three times the interquartile range (IQR) (less than Q1 − (3.0 × IQR) or more than Q3 + (3.0 × IQR)) were considered outliers and excluded from the analysis. Differences in the baseline characteristics between the two groups were tested with the Student’s *t*-test for continuous variables and chi-squared or Fisher’s exact test for categorical variables. Safety and effectiveness comparisons between or within groups were analyzed using a linear mixed-effects model with a random subject effect and fixed effects (group, week, group, the interaction between group and week) after adjustment for covariates. Covariate screening was analyzed using Empower (X&Y Solutions, Inc., Boston, MA, USA). Associations between HF scores or HF severity and bone turnover markers were tested in linear regression models. Data were presented as means and standard errors. All statistical analysis was performed using SAS (version 9.4; SAS Institute, Inc., Cary, NC, USA).

## 3. Results

### 3.1. Baseline Characteristics

Of the 147 subjects recruited, 84 eligible subjects were enrolled and randomized into either the KM or placebo group. Eight subjects dropped out before week 12 due to personal reasons (*n* = 6), taking an omega-3 supplement (*n* = 1), and participating in another clinical trial (*n* = 1). Finally, 76 participants (90%) completed the trial (Figure 1).

Table 1 shows the baseline characteristics of the subjects in ITT analysis. Study groups were well matched, with no significant differences between the KM and placebo groups, demonstrating that the participants were symptomatic peri- or post-menopausal women with a mean age of 51.8 ± 0.4 years, KI of 24.7, and HF score of 31.3. Although the difference in RFS between the two groups was statistically significant (27.0 vs. 23.0), they were all classified as the low fruit/vegetable consumption group [26]. The compliance was excellent in both arms (97.3% vs. 97.5%).

### 3.2. Effect of KM on HFs

Figure 2a–c illustrate the weekly changes in HF scores, severity, and frequency over the study period. These values were calculated based on Web-based digital diaries that record everyday changes. In both groups, all values were reduced substantially from the first week until the end of the study. However, the overall reductions were lower in the KM group than in the placebo group, showing a statistical difference between the two groups in HF severity (*β* = −0.38, *p* = 0.006). As a result, at the end of the study, both the HF scores (*p* = 0.041) and HF severity (*p* < 0.001) were significantly lower in the KM group than in the placebo group. HF scores decreased by 60.1% in the KM group compared with 50.9% in the placebo group from the baseline, while HF severity decreased by 40% in the KM group compared with 26.3% in the placebo group from the baseline. Interestingly, the improvements in the HF scores and HF severity were better at night-time than at daytime, as visualized in the heat map (Figure 2d).

When using the KI and MenQOL retrospective questionnaires, the differences in HF scores’ changes were not statistically significant between the placebo and KM groups. However, the within-group scores were significantly lower after the 12-week intervention than at baseline for both groups, indicating a trend for improving (*p* < 0.05 for all; upper part of Table 2). In the meantime, we determined various safety parameters, including vital signs and hematologic, blood biochemical, and urine analysis. The analysis showed that these values are normal, with no significant differences between the two groups at post-treatment (Appendix A). Besides, we found that there were no significant differences in endometrial thickness and LH level between the two groups. Although the FSH (*β* = −10.81, *p* = 0.037) and estradiol (*β* = 46.23, *p* = 0.043) levels were statistically different, all values remained normal ranges for peri- or post-menopausal women (lower part of Table 2). We note that, the FSH level rises above 40 mIU/mL during menopause [30], while estradiol level decreases from 30 to 400 pg/mL to 0–30 pg/mL [31]. We also note that the changes were opposite in the placebo and KM groups, indicating a potential protective capacity KM to suppress the hormonal changes in menopause. Finally, there were no significant adverse events and no difference in the total number of adverse events between them.

### 3.3. Effect of KM on Bone Turnover Markers

NTx and CTx, the amino- and carboxyterminal cross-linked telopeptides of type I collagen, are two bone resorption markers widely used in the clinical research setting [32]. In this study, the 12-week consumption of KM led to a significant and negative impact on CTx compared with the placebo (*β* = −0.05, *p* = 0.027). A similar result was obtained for NTx (*β* = −0.80), although neither reached statistical significance because of the high variability (upper part of Table 3).

Bone formation markers are synthesized by active osteoblasts and present in the circulation. OC is a hydroxyapatite-binding protein mainly synthesized by osteoblasts, while ALP is a membrane-bound glycoprotein present in four isoforms in the liver, intestine, placenta, and bone. BALP plays a role in osteoid formation and calcification by enzymatic degradation of pyrophosphate, a naturally occurring inhibitor of mineralization [33]. In this study, the KM group showed positive effects for both OC (*β* = 1.03) and BALP (*β* = 0.67) compared with the placebo group, although there were no significant statistical differences in the levels of OC and BALP between the two groups. Moreover, the difference in OC levels between baseline and the end of the study was significantly different after the 12-week consumption of KM (*p* = 0.001; lower part of Table 3).

### 3.4. Associations between HF Scores and Bone Turnover Markers

Associations between HF scores and bone turnover markers were tested using multivariable linear regression. The results showed significant inverse correlations between the changes in HF score and OC (*r* = −0.264, *p* = 0.031), as well as those in HF severity and OC (*r* = −0.228, *p* = 0.056). In parallel with these results, the changes in CTx were positively correlated with those in HF score and HF severity, although they did not reach statistical significance (Figure 3).

## 4. Discussion

The results of this randomized controlled clinical trial involving 84 peri- and post-menopausal women with moderate HFs support the hypothesis that KM might be an effective intervention for achieving reductions in HFs and improvements in bone turnover markers, as compared with the placebo. Another significant outcome of this clinical trial was the verified safety of KM. High-dose consumption of some isoflavones was shown to increase the risks of endometrial hyperplasia because of a high ER-binding affinity [34]. Therefore, we measured endometrial thickness and observed vital signs in all participants to provide precautionary measures and suggestions. As a result, we confirmed no difference in the incidence of adverse events between the two groups. These data provide additional support for the safety of KM use [27,28]. Kim et al. [35] presented an in vitro study highlighting a potential role of kudzu flower extract as an anti-endometriotic agent with inhibition against endometriotic cell adhesion and migration. In a mouse model of menopause, Sternlicht et al. [36] showed that kudzu flower extract exerted a noticeable down-regulation of matrix metalloproteinase, which regulates migration, invasion, and proliferation of various cells.

Although not fully understood, estrogen withdrawal is thought to trigger several neuroendocrine pathways in humans, ultimately causing thermoregulatory dysfunction in the hypothalamus. As the thermoregulatory zone becomes lower and narrower in menopausal women, a small change in core body temperature may trigger the heat loss mechanisms, leading to HF symptoms [37]. In the present study, the KM and placebo groups showed a 60.1% and 50.9% reduction in HF score and a 40.0% and 26.3% reduction in HF severity, respectively, from baseline to the end of treatment. These rates of reduction in HF score and severity are comparable or better than those achieved through non-hormonal interventions, including soy isoflavones, black cohosh, and red clover [38]. It is reported that a mixture of tectorigenin derivatives dramatically reduces the OVX-induced rise in tail skin temperature in mice [17]. Tectorigenin derivatives are unique isoflavones found in kudzu flower but not in soy [39]. Hence, KM’s potential as an effective non-hormonal alternative for the management of menopausal HFs might be attributed to the presence of tectorigenin derivatives. Additionally, we extended our findings to compare the impact of KM on the day and night HF outcomes using a heat map. Results showed a better improvement in night-time HFs than daytime HFs. This observation suggests that KM might also improve sleep quality in individuals with night-time HFs but needs further study.

The measurement of HFs can present unique methodologic challenges due to intangible and subjective characteristics [27]. Traditional retrospective summative HF diaries are inaccurate measures of actual HF status because of the subject’s memory capacity limitations and the influence of mood on symptom reporting [28]. We thus decided to use prospective electronic digital diaries over 12 weeks from 7 days before treatment. This approach is not only a cost-effective data acquisition scheme but also relatively more precise in quantifying HFs because it reduces the limitations of recall [28]. Furthermore, the current study supplemented the severity and frequency data of menopausal symptoms with the self-administered questionnaires to monitor quality-of-life as a treatment outcome. Although the results demonstrated similar improvements, there were no significant differences between the two groups. The failure of statistical significance is likely due to both the less frequent measurement intervals and the higher dependence on subjective perception compared with the electronic digital diaries. Moreover, although validated, the KI and MenQOL have weaknesses, such as limited focus and lack of psychometric assessments [40].

The onset of menopausal HFs is associated with various adverse disease risk factors [41]. In the current study, we explored the associations between HFs and bone turnover markers because estrogen deficiency in menopause is related to rapid bone loss [42]. Notably, the changes in HFs were positively correlated with those in CTx and negatively correlated with those in OC. That is, there seemed to be higher bone resorption and lower bone formation among subjects with more HFs. These results were supported by earlier animal studies where KM, tectorigenin derivatives, and hesperidin were all effective in down-regulating receptor activator of nuclear factor κB ligand (RANKL) in OVX mice [17,18]. RANKL is a transmembrane ligand expressed on osteoblasts, which activates RANK in osteoclasts and triggers osteoclast maturation and bone resorption [43]. Other studies also indicated the inhibitory activities of tectorigenin derivatives against RANKL-induced osteoclastogenesis [44,45], and a preventive role of hesperidin against bone loss in OVX animals [16,46]. However, mixed results have been reported regarding the association between HFs and BMD in humans. Salamone et al. [47] observed a noticeably lowered BMD in the spine, hip, and whole body in 290 women reporting HFs. Similarly, Lee and Kanis [48] showed a positive association between VMS and vertebral fractures. In contrast, Scoutellas et al. [49] found no association between VMS and vertebral fractures in post-menopausal women aged 50–64 years. It was suggested that these conflicting results might originate from differences in subject ages, use of hormone therapy, or recall period [50].

Limitations of this study include that we did not measure levels of serum serotonin and epinephrine, which are thought to play an important role in thermoregulatory function. We also did not measure the levels of RANKL gene expression responsible for the bone turnover function. As a result, direct mechanisms of action of KM in relieving HFs and bone turnover were not fully understood. Nonetheless, this randomized controlled trial provides a novel finding that KM appears to alleviate HFs during the menopausal transition. The KM was well-tolerated in this study. Besides, this study had many strengths. First, we included peri- and post-menopausal women because of the peak in HF severity during the latter part of the menopausal transition near the final menstrual period [51]. Second, HF data were collected prospectively using electronic digital diaries, and daytime and night-time HFs were also analyzed separately. Third, the intervention lasted for 12 weeks, which is a relatively longer follow-up than most of the other studies [52]. Lastly, KM has been authenticated by chemical analysis of three tectorigenin derivatives and hesperidin.

## 5. Conclusions

In conclusion, given the positive effects of KM on HFs and bone turnover and the favorable side effect profile, supplementation with a KM dose of 1150 mg/day seems to be an acceptable option for reducing HF symptoms, as well as improving bone turnover, compared with the placebo group, during the menopausal transition. Future studies are needed to investigate the underlying mechanisms behind the effects and synergistic interactions between tectorigenin derivatives and hesperidin on HFs and bone turnover.

## Figures and Tables

**Figure 1 nutrients-12-03237-f001:**
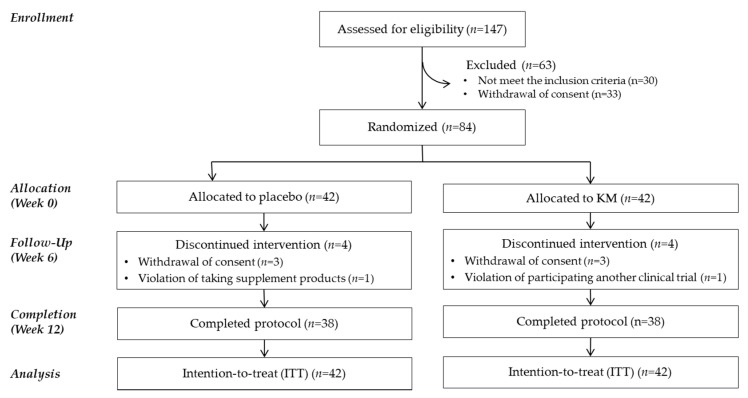
CONSORT flow diagram of the study. CONSORT, Consolidated Standards of Reporting Trials; KM, an extract mixture of kudzu flower and mandarin peel.

**Figure 2 nutrients-12-03237-f002:**
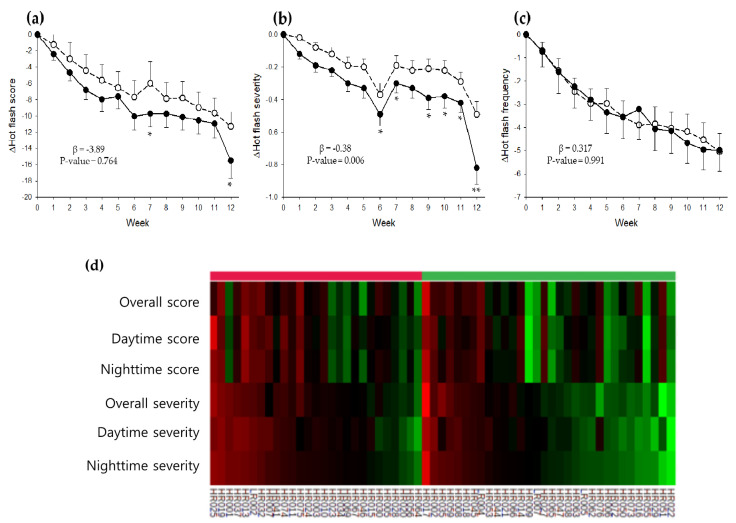
Changes in mean HF score (**a**), severity (**b**), and frequency (**c**) from baseline over the 12-week study period. Dots are experimental points (KM: black circles, placebo: empty circles). Heat map representing the changes in HF score and severity from baseline to the end of week 12 (**d**). The HF scores were calculated by multiplying the frequency experienced at that severity level. Then, the resulting points were summed to give a total daily score. Each daily score was averaged per week. Each value represents the mean + SE. Each column of the map represents one subject with all data for HF score and severity at total, daytime, and night-time in the heatmap. KM, an extract mixture of kudzu flower and mandarin peel; HF, hot flash. The *p*-values were obtained from a linear mixed-effect model adjusted for covariates. * *p* < 0.05, ** *p* < 0.001.

**Figure 3 nutrients-12-03237-f003:**
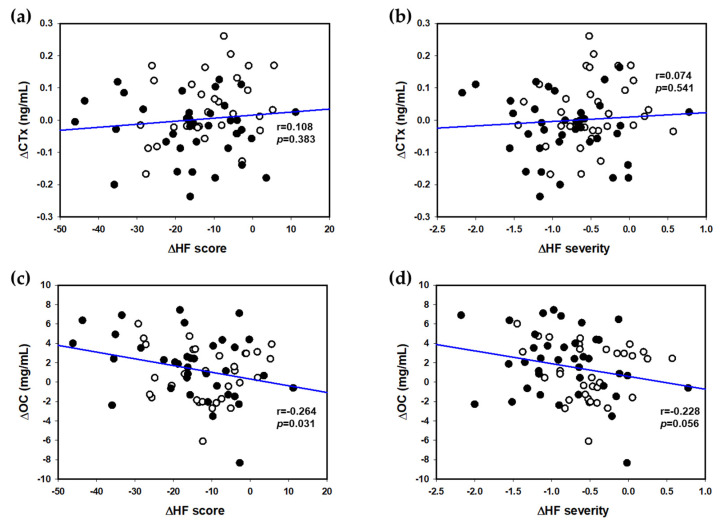
Associations between differential scores and bone turnover markers. CTx was positively correlated with HF scores (**a**) and HF severity (**b**), while OC was negatively correlated with HF scores (**c**) and HF severity (**d**). Dots are experimental points (KM: black circles, placebo: empty circles), and the continuous line indicates the straight-line fit by linear regression. KM, an extract mixture of kudzu flower and mandarin peel; HF, hot flashes; CTx, cross-linked C-telopeptides of bone collagen; OC, osteocalcin.

**Table 1 nutrients-12-03237-t001:** Baseline characteristics of subjects in the intention-to-treat analysis.

Variables	Placebo (*n* = 42)	KM (*n* = 42)	*p*-Value
Age (years)	52.0 ± 0.6	51.5 ± 0.5	0.526
Amenorrhea period (months)	29.8 ± 4.5	20.7 ± 2.9	0.096
Hot flash score	34.8 ± 6.7	27.8 ±3.3	0.354
Kupperman Index	25.2 ± 1.3	24.1 ± 1.3	0.532
Alcohol drinker (Y/N)	21/21	19/23	0.662
Alcohol amount (g/week)	8.0 ± 2.2	6.5 ± 1.8	0.598
Smoker (Y/N)	0/42	1/41	1.000
Smoking amount (cigarettes/day)	0.0 ± 0.0	0.1 ± 0.1	0.183
Body weight (kg)	59.8 ± 1.5	57.7 ± 1.2	0.268
BMI (kg/m^2^)	23.7 ± 0.5	23.1 ± 0.5	0.352
Waist circumference (cm)	83.0 ± 1.6	79.1 ± 1.3	0.059
Physical activity (MET-min/week)	1499.5 ± 148.7	1905.2 ± 251.7	0.170
Vigorous activity (h/week)	0.4 ± 0.2	0.2 ± 0.1	0.435
RFS	23.0 ± 1.2	27.0 ± 1.3	0.028
SBP (mmHg)	118.2 ± 1.6	118.2 ± 2.1	0.979
DBP (mmHg)	73.2 ± 1.4	72.7 ± 1.6	0.807

Values are presented as mean ± SE or *n*. Student’s *t*-test for continuous variables and chi-square or Fisher’s exact test for categorical variables were used to compare the difference between the treatments. KM, an extract mixture of kudzu flower and mandarin peel; BMI, body mass index; RFS, recommended food score; SBP, systolic blood pressure; DBP, diastolic blood pressure.

**Table 2 nutrients-12-03237-t002:** Changes in retrospective HF scores and safety parameters.

Variables		Placebo (*n* = 42)	KM (*n* = 42)	Estimate	*p*-Value
Retrospective HF scores					
KI	Week 0	6.86 ± 0.44	5.24 ± 0.37	0.83	0.191
	Week 12	4.63 ± 0.47	4.21 ± 0.4		
	*p*-value	<0.001	0.013		
MenQOL score	Week 0	15.17 ± 0.7	13.64 ± 0.64	0.555	0.163
	Week 12	9.97 ± 0.79	10.26 ± 0.75		
	*p*-value	<0.001	<0.001		
Safety parameters					
Endometrial thickness (mm)	Week 0	5.4 ± 0.4	5.4 ± 0.6	0.52	0.380
	Week 12	5.0 ± 0.4	5.6 ± 0.5		
	*p*-value	0.14	0.793		
FSH (mIU/mL)	Week 0	63.4 ± 5.8	71.0 ± 6.5	−10.81	0.037
	Week 12	68.4 ± 5.5	66.2 ± 7.5		
	*p*-value	0.062	0.271		
Estradiol (pg/mL)	Week 0	42.1 ± 12.8	37.5 ± 11.6	46.23	0.043
	Week 12	23.6 ± 8.4	67.8 ± 19.1		
	*p*-value	0.192	0.115		
LH (mIU/mL)	Week 0	34.7 ± 2.8	35.9 ± 3.1	−0.09	0.986
	Week 12	38.3 ± 2.7	40.1 ± 4.5		
	*p*-value	0.27	0.267		

Values are presented as mean ± SE. Estimates and *p*-values were obtained from a linear mixed-effect model adjusted for covariates. KI, kupperman index; KM, an extract mixture of kudzu flower and mandarin peel; HF, hot flash; MenQOL, menopause quality of life scale; FSH, follicle stimulating hormone; LH, luteinizing hormone.

**Table 3 nutrients-12-03237-t003:** Changes in bone turnover markers in the blood.

Variables		Placebo (*n* = 42)	KM (*n* = 42)	Estimate	*p*-Value
Markers of bone resorption					
CTx (ng/mL)	Week 0	0.43 ± 0.03	0.43 ± 0.03	−0.05	0.027
	Week 12	0.47 ± 0.03	0.39 ± 0.03		
	*p*-value	0.098	0.134		
NTx (ng/mL)	Week 0	3.50 ± 0.32	3.94 ± 0.45	−0.80	0.25
	Week 12	3.53 ± 0.36	3.06 ± 0.23		
	*p*-value	0.922	0.124		
Markers of bone formation					
OC (mg/mL)	Week 0	18.35 ± 0.92	17.64 ± 0.98	1.03	0.17
	Week 12	19.49 ± 0.87	19.48 ± 1.12		
	*p*-value	0.124	0.001		
BALP (mg/mL)	Week 0	15.21 ± 0.73	13.48 ± 0.73	0.67	0.276
	Week 12	14.73 ± 0.70	13.53 ± 0.84		
	*p*-value	0.443	0.435		

Values are presented as mean ± SE. Estimate and p-value were obtained from a linear mixed-effect model adjusted for covariates. KM, an extract mixture of kudzu flower and mandarin peel; CTx, cross-linked C-telopeptides of bone collagen; NTx, cross-linked N-telopeptide of bone collagen; OC, osteocalcin; BALP, bone-specific alkaline phosphatase.

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
