# Peer review of "Efficacy and Safety of Kudzu Flower–Mandarin Peel on Hot Flashes and Bone Markers in Women during the Menopausal Transition: A Randomized Controlled Trial"

_nutrients, 2020, doi:10.3390/nu12113237_

Round 1

Reviewer 1 Report

This is a report of a double-blind randomized controlled trial investigating the effects of a mixture of kudzu flower and mandarin peel on vasomotor symptoms and bone metabolism markers in middle-aged Korean women.

The study was well-designed, the trial was appropriately performed, and the paper was beautifully written.

The reviewer has just a single point to raise. The authors claimed that the mixture was safe, concerning endogenous hormone levels, endometrial thickness, and so on, which data were not shown in the manuscript. The pre-post changes in these parameters in each group should be presented to prove that there are no potential harms caused by the product.

The reviewer strongly recommends this paper should be transferred to the special edition in the same journal “Nutrition Challenges for Middle-Aged and Older Women” according to the specificity of the topic of the current work.

Author Response

Response to Reviewer 1 Comments

Dear Editor,

We want to thank Editor and Reviewers for taking the time and effort necessary to review the manuscript. We sincerely appreciate all the valuable comments and suggestions, which helped us improve the manuscript's quality. We have made many changes in our manuscript to incorporate the questions and suggestions as thoroughly as possible. We have summarized our response below point-by-point, and the corrections have been marked by red color in the revised manuscript.

Kind regards,

Oran Kwon

The authors claimed that the mixture was safe, concerning endogenous hormone levels, endometrial thickness, and so on, which data were not shown in the manuscript. The pre-post changes in these parameters in each group should be presented to prove that there are no potential harms caused by the product.

→ Thank you for this critical comment. We have presented the data on the endogenous hormone levels and endometrial thickness by extending Table 2. We have also added texts to describe the findings (line 224-234). We hope that this new version is satisfactory. Besides, we provided the data on various safety parameters, including vital signs and hematologic and blood biochemical analysis in Supplementary Table 1.

Reviewer 2 Report

Efficacy and Safety of Kudzu Flower–Mandarin Peel on Hot Flashes and Bone Markers in Women During the Menopausal Transition: A Randomized Controlled Trial

This RCT paper discusses the effect of Kudzu Flower–Mandarin Peel on Hot Flashes and Bone Markers among menopausal women. The paper will be benefited from addressing the following feedback.

Abstract

  • Make sure to clarify the ratio of experimental to placebo group whether it is 1:1 or 1:2.

Introduction

  • page 2, line 46: Please cite consistently- cite the web
  • page 2, line 51-58: although it si good that they physiological effect of these plants are described, the timing should also be mentioned. This will help us to determine the minimum follow up time for the experimental and control groups in the RCT.
  • Page 2, line 59-64: the findings of both studies should be presented here. If they show efficacy then it would be good to go for this study. However, if these studies showed none, then study should be ‘tested’ in mice before experimented in human.
  • I would also suggest whether other literature are there on the area or mention clearly that there is no study on the issue among these population

Methods

  • Page 2, line 71-79: please mention if these products received a licence from the US FDA.
  • Page 3, line 104: clarify the experimental to placebo ratio, and mention how many women participated in each group.
  • Why 84? How was the sample size calculated? Some information in lines 156-158 of page 3 are mentioned but this should be brought to a separate section detailing the sampling process including the sample size determination and recruitment strategy.
  • Page 3, line 106: could you add a justification why 12-weeks timeline is enough/appropriate for the experiment?
  • Page 3, line 110-112, and line 130-134: please add citations for the tools.
  • Page 3, line 155: any non-parametric inferential analysis?

Results

  • Written very well and in detail – no comment
  • The contrast of figures is not clear—please improve the resolution
  • Page 3, line 218…please add in a bracket (whether lower means good or bad)

Discussion

  • Page 9, line 324 onwards: Limitation, recruitment bias as some may do not have access to posters or online advertisements. Also other limitations related to the tools?
  • DO you have any suggestion for the women excluded from this study? There are women on hormone therapy, for example, who may be affected by the HF.

Conclusion

  • None

Author Response

Response to Reviewer 2 Comments

Dear Editor,

We want to thank Editor and Reviewers for taking the time and effort necessary to review the manuscript. We sincerely appreciate all the valuable comments and suggestions, which helped us improve the manuscript's quality. We have made many changes in our manuscript to incorporate the questions and suggestions as thoroughly as possible. We have summarized our response below point-by-point, and the corrections have been marked by red color in the revised manuscript.

Kind regards,

Oran Kwon

Reviewer #2

Abstract: Make sure to clarify the ratio of experimental to placebo group whether it is 1:1 or 1:2.

→ Corrected as suggested, which reads as follows: "randomly assigned in a 1:1 ratio" (line 15). Thank you.

Introduction: page 2, line 46: Please cite consistently- cite the web page 2 line 51-58

→We cited the web page similarly to other references (line 45; reference #12).

Introduction: although it is good that they physiological effect of these plants are described, the timing should also be mentioned. This will help us to determine the minimum follow up time for the experimental and control groups in the RCT.

→ Following the reviewer's comment, we have added the previous two animal studies (line 59 and 62).

Introduction: Page 2, line 59-64: the findings of both studies should be presented here. If they show efficacy then it would be good to go for this study. However, if these studies showed none, then study should be 'tested' in mice before experimented in human. I would also suggest whether other literature are there on the area or mention clearly that there is no study on the issue among these population.

→ Thank you for your helpful comments. We have revised the introduction section accordingly, which reads as follows: "However, to the best of our knowledge, KM's protective attributes have not been confirmed in human studies. It triggers us to study the potentials of KM in human subjects." (line 64-65).

Methods: Page 2, line 71-79: please mention if these products received a licence from the US FDA.

→ Thank you for the valuable comment. We have added the following sentences; "The mandarin peel and kudzu flower are listed as Old Dietary Ingredients marketed in the US before October 15, 1994”. Therefore, they might be used legally in a dietary supplement without submitting the Food and Drug Administration notification." (line 72-74)  

Methods: Page 3, line 104: clarify the experimental to placebo ratio, and mention how many women participated in each group.

→ Corrected as suggested, which reads as follows: "fixed-dose of KM (1,150 mg) or matching placebo in a ratio of 1:1. A total of 84 eligible subjects were randomized to the KM (n = 42) or placebo (n = 42) groups" (line 110-111). Thank you.

Why 84? How was the sample size calculated? Some information in lines 156-158 of page 3 are mentioned but this should be brought to a separate section detailing the sampling process including the sample size determination and recruitment strategy.

→ We have moved the power calculation from the statistics to the subjects section to provide information about the sample size determination and recruitment strategy in one place (line 85-87).

Page 3, line 106: could you add a justification why 12-weeks timeline is enough/appropriate for the experiment?

→ We revised the text to justify the duration of a 12-week timeline of this study (line 113-115). Thank you.

Page 3, line 110-112, and line 130-134: please add citations for the tools.

→ Corrected as suggested (line 118, 120, and 140; references #24, #25, #26, and #29).

Page 3, line 155: any non-parametric inferential analysis?

→ We appreciate the reviewer's comment. The adverse events were monitored throughout the study. We clarified that the adverse events were monitored throughout the study method (line 162). We also revised the results to provide safety information in one place (line 233-234). 

Results: The contrast of the figures is not clear—please improve the resolution.

→ We have improved the resolution of figures.

Results: Page 3, line 218…please add in a bracket (whether lower means good or bad)

→ We revised the text to explain that lower means good, which reads as follows: "indicating a trend for improving" (line 223).

Discussion: Page 9, line 324 onwards: Limitation, recruitment bias as some may do not have access to posters or online advertisements. Also other limitations related to the tools?

→ Thank you for giving us another chance to think about the limitation of our study. However, we do not have further suggestions related to recruitment and tools.  

DO you have any suggestion for the women excluded from this study? There are women on hormone therapy, for example, who may be affected by the HF.

→ We concur with the reviewer's comment. However, the women on hormone therapy were excluded to avoid confounding in study design.

Round 2

Reviewer 2 Report

It would be great if some one with native English read the paper again to improve the readability.